# Impact of Acetate versus Citrate Dialysates on Intermediary Metabolism—A Targeted Metabolomics Approach

**DOI:** 10.3390/ijms231911693

**Published:** 2022-10-02

**Authors:** José Jesús Broseta, Marta Roca, Diana Rodríguez-Espinosa, Luis Carlos López-Romero, Aina Gómez-Bori, Elena Cuadrado-Payán, Ramón Devesa-Such, Amparo Soldevila, Sergio Bea-Granell, Pilar Sánchez-Pérez, Julio Hernández-Jaras

**Affiliations:** 1Department of Nephrology and Renal Transplantation, Hospital Clínic of Barcelona, 08036 Barcelona, Spain; 2Analytical Unit Platform, Medical Research Institute Hospital La Fe (IIS La Fe), 46026 Valencia, Spain; 3Department of Nephrology, Hospital Universitari i Politècnic La Fe, 46026 Valencia, Spain; 4Department of Nephrology, Consorci Hospital General Universitari de València, 46014 Valencia, Spain

**Keywords:** hemodialysis, targeted metabolomics, dialysate, citrate, acetate, acetate-free

## Abstract

Acetate is widely used as a dialysate buffer to avoid the precipitation of bicarbonate salts. However, even at low concentrations that wouldn’t surpass the metabolic capacity of the Krebs tricarboxylic acid (TCA) cycle, other metabolic routes are activated, leading to undesirable clinical consequences by poorly understood mechanisms. This study aims to add information that could biologically explain the clinical improvements found in patients using citrate dialysate. A unicentric, cross-over, prospective targeted metabolomics study was designed to analyze the differences between two dialysates, one containing 4 mmol/L of acetate (AD) and the other 1 mmol/L of citrate (CD). Fifteen metabolites were studied to investigate changes induced in the TCA cycle, glycolysis, anaerobic metabolism, ketone bodies, and triglyceride and aminoacidic metabolism. Twenty-one patients completed the study. Citrate increased during the dialysis sessions when CD was used, without surpassing normal values. Other differences found in the next TCA cycle steps showed an increased substrate accumulation when using AD. While lactate decreased, pyruvate remained stable, and ketogenesis was boosted during dialysis. Acetylcarnitine and myo-inositol were reduced during dialysis, while glycerol remained constant. Lastly, glutamate and glutarate decreased due to the inhibition of amino acidic degradation. This study raises new hypotheses that need further investigation to understand better the biochemical processes that dialysis and the different dialysate buffers induce in the patient’s metabolism.

## 1. Introduction

Chronic kidney disease (CKD) leads to the dysfunction of various metabolic systems and the accumulation of the so-called uremic toxins [1]. In dialysis-dependent CKD (DD-CKD) patients, several metabolites have been correlated with cardiovascular disease, inflammatory parameters, disease progression, nutritional status, cognitive function, hypoxia and oxidative stress, body mass wasting, and even sleep cycle disturbance [2,3,4,5,6]. Dialysis by itself may worsen several metabolic functions [1].

One of the most challenging aspects of dialysate production is allowing the coexistence of a solution of divalent cations and bicarbonate without precipitation [7]. Acetate is widely used as a weak acid acting as a dialysate buffer, thus avoiding this precipitation. Even at the low concentrations (3–4 mmol/L) used, it reaches blood levels higher than physiological [8], leading to undesirable outcomes by poorly understood mechanisms.

Among the several compounds proposed as alternatives to acetate, citrate is important [7], as it has been associated with improved hemodynamic tolerance and inflammatory biomarkers in DD-CKD patients [9,10,11,12,13]. Even though the concentration in the dialysate is five times lower than that needed for anticoagulation, citrate also allows patients to receive a reduced heparin dose without increasing the risk of filter coagulation [14,15,16,17]. Using acetate and citrate as dialysate buffers seems logical, since they would be later mainly transformed into bicarbonate through the TCA cycle. While citrate would enter directly, acetate would rapidly enter the cell, bind to coenzyme A to form acetyl-CoA, and then be oxidized in the TCA cycle, acting both as potentially safe and cheap bicarbonate precursors [18].

In this paper, we analyzed the inter-dialysis and intra-dialysis changes in the metabolomic profile of hemodialysis patients after being exposed to either citrate or acetate dialysates using a targeted metabolomics approach. Thus, we aim to add information that could help biologically explain the clinical improvements found in patients using citrate dialysate [11,19,20,21].

## 2. Results

### 2.1. Participants

Twenty-one patients with a mean age of 62.25 ± 13.77 years were included. Ten (48%) were male. Two (9.5%) were on high-flux hemodialysis and nineteen (90.5%) on post-dilution online hemodiafiltration. Twelve (57.1%) were dialyzed through an arteriovenous fistula and nine (42.9%) through a tunneled catheter.

### 2.2. Metabolomic Analysis

The concentrations of each metabolite in the four measured times are represented in Table 1. Normalized data by Z-scoring are also illustrated in a heat map (Figure 1).

#### 2.2.1. Acetylcarnitine

There was a statistically significant difference in acetylcarnitine concentrations among the four measurements, χ2(3) = 49.96, *p* < 0.0001 (Figure 2). A decrease appeared between pre- and post-dialysis levels using AD (*Z* = −4.015, *p* < 0.0001) and CD (*Z* = −3.98, *p* < 0.0001); however, there was no significant difference between AD and CD either in pre- (*Z* = −1.303, *p* = 0.192) or post-dialysis (*Z* = −2.461, *p* = 0.014).

#### 2.2.2. Fumarate

There was a statistically significant difference in fumarate concentrations among the four measurements, χ2(3) = 12.714, *p* = 0.005 (Figure 2). Its levels remained stable during the dialysis session with both AD (*Z* = −0.434, *p* = 0.664) and CD (*Z* = −0.017, *p* = 0.986). Pre- (*Z* = −2.555, *p* = 0.011) and post-dialysis (*Z* = −3.007, *p* = 0.003) levels were significantly different.

#### 2.2.3. Citrate

Mean citrate concentrations significantly differed between the four measurements [*F*(2.816, 56.326) = 119.533, *p* < 0.0001] and consistently in the pairwise comparisons (Figure 2). During the sessions, citrate decreased [10,778.19 (95% CI, 7147.642–14,408.738), *p* < 0.0001] with AD and increased with CD [−17,038.96 (95% CI, −20,739.052–−13,338.871), *p* < 0.0001]. Pre- [5617.231 (95% CI, 1538.828–9695.635), *p* = 0.004] and post-dialysis [−22,199.92 (95% CI, −25,691.342–−18,708.499), *p* < 0.0001] levels significantly differed, being higher in the pre-dialysis AD measurement but higher in the post-dialysis treatment after one session using CD.

#### 2.2.4. Isocitrate

There was a statistically significant difference in isocitrate concentrations among the four measurements, χ2(3) = 52.029, *p* < 0.0001. Its levels were significantly reduced during dialysis with both AD (*Z* = −4.015, *p* < 0.0001) and CD (*Z* = −4.015, *p* < 0.0001). Moreover, the pre- (*Z* = −2.763, *p* = 0.006) and post-dialysis (*Z* = −3.285, *p* = 0.001) significantly differed.

#### 2.2.5. Myo-Inositol

There was a statistically significant difference in myo-inositol concentrations among the four measurements, χ2(3) = 51.641, *p* < 0.0001. Its levels were significantly reduced during dialysis with both AD (*Z* = −4.015, *p* < 0.0001) and CD (*Z* = −4.015, *p* < 0.0001). However, there were no pre- (*Z* = −1.13, *p* = 0.259) or post-dialysis (*Z* = −1.207, *p* = 0.027) significant differences.

#### 2.2.6. Glutamate

There was a statistically significant difference in glutamate concentrations among the four measurements, χ2(3) = 14.257, *p* = 0.003. During the sessions, its levels increased. These differences were statistically significant when using CD (*Z* = −3.493, *p* < 0.0001) but not when using AD (*Z* = −2.068, *p* = 0.039). Neither, pre- (*Z* = −1.199, *p* = 0.23), nor post-dialysis (*Z* = −0.643, *p* = 0.52) results significantly differed (Figure 2).

#### 2.2.7. Pyruvate

There were no statistically significant differences in pyruvate concentrations among measurements, χ2(3) = 4.429, *p* = 0.219 (Figure 2).

#### 2.2.8. Malate

There was a statistically significant difference in malate concentrations among the four measurements, χ2(3) = 17.857, *p* < 0.0001 (Figure 2). Its levels did not change because of the dialysis session with AD (*Z* = −0.713, *p* = 0.476) or CD (*Z* = −0.713, *p* = 0.476). Pre- (*Z* = −3.285, *p* = 0.001) and post-dialysis (*Z* = −3.007, *p* = 0.003) levels were significantly higher with AD.

#### 2.2.9. Glycerol

There was a statistically significant difference in glycerol concentrations among the four measurements, χ2(3) = 11.842, *p* = 0.008 (Figure 2); however, in pairwise comparison, there was neither a significant change during dialysis with AD (*Z* = −1.826, *p* = 0.068) or CD (*Z* = −2.201, *p* = 0.028), nor between pre- (*Z* = −0.77, *p* = 0.441) or post-dialysis (*Z* = 0, *p* = 1) results.

#### 2.2.10. Lactate

Mean lactate concentrations differed between the four measurements (*F*(2.478, 49.568) = 22.063, *p* < 0.0001) (Figure 2). Lactate levels decreased after dialysis both with AD (83633.688 (95% CI, 26,068.803–141,198.572), *p* = 0.002) and CD (99,583.827 (95% CI, 51,404.704–147,762.949), *p* < 0.0001). Pre-dialysis levels did not differ (47,432.131 (95% CI, −17,205.387–112,069.648), *p* = 0.265) but post-dialysis levels did (63,382.270 (95% CI, 20,177.688–106,586.852), *p* = 0.002), with lower levels after a dialysis with CD.

#### 2.2.11. 2-Oxoglutarate

There was a statistically significant difference in 2-oxoglutarate concentrations among the four measurements, χ2(3) = 16.371, *p* = 0.001 (Figure 2). Its levels did not significantly change during the dialysis session with AD (*Z* = −1.79, *p* = 0.073) or CD (*Z* = −2.242, *p* = 0.025). No differences occurred between pre-dialysis levels (*Z* = −1.721, *p* = 0.085), but post-dialysis (*Z* = −2.902, *p* = 0.004) values were found to be higher after a session with CD.

#### 2.2.12. Acetoacetate

There was a statistically significant difference in acetoacetate concentrations among the four measurements, χ2(3) = 37.057, *p* < 0.0001 (Figure 2). During the sessions, both AD (*Z* = −3.91, *p* < 0.0001) and CD (*Z* = −3.18, *p* = 0.001) induced significant increases. Pre- (*Z* = −2.833, *p* = 0.005), but not post-dialysis (*Z* = −1.581, *p* = 0.114) levels differed between dialysates, being higher in the CD measurement.

#### 2.2.13. Succinate

There was a statistically significant difference in succinate concentrations among the four measurements, χ2(3) = 34.314, *p* < 0.0001 (Figure 2). During the dialysis sessions, succinate levels increased both with AD (*Z* = −3.771, *p* < 0.0001) and CD (*Z* = 3.806, *p* < 0.0001), but there were no pre- (*Z* = −1.06, *p* = 0.289) or post-dialysis (*Z* = −1.581, *p* = 0.114) differences.

#### 2.2.14. 3-hydroxybutyrate

There was a statistically significant difference in 3-hydroxybutyrate concentrations among the four measurements, χ2(3) = 33.089, *p* < 0.0001 (Figure 2). Its levels significantly increased during the sessions with AD (*Z* = −3.509, *p* < 0.0001) and CD (*Z* = −3.733, *p* < 0.0001), but there were no differences between pre- (*Z* = −1.761, *p* = 0.078) or post-dialysis (*Z* = −1.408, *p* = 0.159) levels.

#### 2.2.15. Glutarate

There was a statistically significant difference in glutarate concentrations among the four measurements, χ2(3) = 33.857, *p* < 0.0001 (Figure 2). During the session, glutarate levels were reduced with both AD (*Z* = −3.91, *p* < 0.0001) and CD (*Z* = −3.632, *p* < 0.0001), but no significant differences were found between pre- (*Z* = −0.122, *p* = 0.903) and post-dialysis (*Z* = −1.025, *p* = 0.305) levels.

## 3. Discussion

Metabolomics opens the door to studying human metabolism in response to dialysis-induced changes in intermediary metabolism. This study, performed by a targeted metabolomic approach, found noteworthy differences between dialysates that raise new hypotheses that should be further investigated to explain the deleterious acetate effects on dialysis patients [9,10,11,12,13]. Acetyl-CoA levels exceeded by 20–70 times the physiological level despite the apparently small concentrations of acetate in the AD [8,22,23,24]. Although they do not seem to surpass the TCA cycle maximum acetyl-CoA rate of metabolism (diminished from 300 mmol/h in healthy adults to 150 mmol/h in DD-CKD patients [23,25]), acetyl-CoA enters other metabolic routes triggering metabolism disturbances [7,24].

### 3.1. First Carbon Oxidation Pathway of the TCA Cycle-Related Metabolites

It is noteworthy that after twelve sessions under AD but not under CD, patients had elevated pre-dialysis citrate levels linked to chronic accumulation of acetyl-CoA. This difference reverted during dialysis since citrate rose in blood when CD was used, whereas it decreased when AD was used. Nonetheless, this is inconsistent with other published papers in which it remained stable [8,18,24]. Citrate physiological levels are reasonably similar to our four measurements [26,27], and even after one dialysis with CD did not surpass the upper limit of the proposed normal range. Hence, a supraphysiological accumulation of citrate would not occur due to an improved balance between citrate generation and loss by the dialyzer than with acetate [22,24], in contrast to the 20 to 70 times higher acetate levels observed with AD.

Citrate is isomerized to isocitrate, whose levels decreased more intensely during the sessions with AD. Moreover, pre-dialysis levels differed, so this difference was maintained in the long term. The results observed in all study conditions were considerably lower than physiological [26].

The next step was measurement of 2-oxoglutarate, a rate-determining intermediate in the TCA cycle [28]. 2-oxoglutarate levels were 3–4 times supraphysiological in all study situations [26], with a non-significant increase during dialysis sessions, though post-dialysis levels became significantly higher with CD. This finding, not found in previous reports [18], would have short-term impact.

### 3.2. Second Carbon Oxidation Pathway of the TCA-Cycle Related Metabolites

Succinate levels rise during the dialysis sessions, which was due to the treatment itself, as no differences between dialysates were found; however, they were lower than physiological levels in all study situations. Fumarate, a metabolite that also takes part in several other pathways, was not modified during the dialysis sessions but had higher pre- and post-dialysis levels when patients were under AD, reaching levels 10 times higher than those seen in healthy adults. Consistent with previously published data, malate remained stable during the dialysis sessions and close to physiological levels [18,26,29].

These changes in the TCA cycle metabolites should not be considered trivial. Acetyl-CoA, succinate, and fumarate can alter immune system responses [28]. Succinate, malate, and fumarate promote tumorigenesis due to their ability to inhibit prolyl hydroxylase-containing enzymes [30]. Lymphangiogenesis and the maintenance of stem cells pluripotency have been associated with acetyl-CoA and 2-oxoglutarate, respectively. The latter is also related to decreased protein catabolism, increased protein synthesis, and increased circulating plasma levels of hormones such as insulin, growth hormone, and IGF−1 [28]. Furthermore, it plays a role in CD4+ T-cell differentiation [31].

### 3.3. Glycolysis and Anaerobic Metabolism-Related Metabolites

Glycolysis has three irreversible steps mediated by phosphofructokinase, hexokinase, and pyruvate kinase enzymes. Citrate inhibits phosphofructokinase and hexokinase, while the pyruvate kinase is hampered both by citrate and acetyl-CoA. Therefore, both dialysates may play a role in glycolysis inhibition. We found no differences in pyruvate levels during the dialysis session nor between dialysates, as previously reported [8,18]. Pyruvate represents the sole metabolic pathway of lactate, with a bidirectional conversion between the two anions activated by lactate dehydrogenase. During dialysis, acetate displaces pyruvate as a precursor of acetyl-CoA, which, coupled with its generation by glycolysis and the degradation of oxalacetate by the action of pyruvate carboxylase into pyruvate (accumulated by increased acetyl-CoA metabolism in the TCA cycle), may increase its levels [24]. Lactate is the predominant organic acid produced during dialysis, resulting in inefficient use of metabolic substrates for energy production and an irreversible loss of alkali, contributing to the increase of pCO_2_ [32]. Moreover, the carbonization and decarboxylation of an acetyl-CoA molecule in the TCA cycle is favorable to oxidation and displaces the oxide-reductant pairs increasing pyruvate levels at the expense of lactate [24]. Even with this overproduction, its levels were reduced during dialysis, particularly when using CD, but were higher than the physiological range in all study situations. Therefore, comparing the lactate/pyruvate ratio is interesting. It was higher in AD pre-dialysis measurements and increased with AD while decreasing with CD during the session. These results are consistent with prior reports [8,24]. Myo-inositol levels decreased during dialysis, but as a uremic toxin dialyzed by the membranes, the absence of differences does not provide information of interest.

### 3.4. Lipidic Metabolism-Related Metabolites

The decline of Acetylcarnitine levels during dialysis may not be due to activation of β-oxidation of fatty acids but losses through membranes. The lack of acetylcarnitine in DD-CKD patients has been related to cardiac complications, impaired functional capacities, symptomatic intradialytic hypotension, and erythropoietin-resistant anemia [33]. Its role in various neurodegenerative processes has also been hypothesized [34]. However, pre-dialysis levels in this study were higher than that of healthy adults [29]. Glycerol levels did not change, so no fatty acid release seemed to occur; however, some measurements were under the LQ, representing a potential bias for interpretation. Glycerol levels were far below physiological levels, despite enormous heterogeneity [29,35].

High levels of acetyl-CoA can combine into acetoacetyl-CoA, essential for synthesizing cholesterol and ketone bodies. 3-hydroxybutyrate levels increased during the dialysis treatment irrespective of the dialysate, indicating dialysis-induced lipolysis [32] and consistent with previous reports stating that it is the second predominant organic acid produced during dialysis [8,24,32]. AD pre-dialysis measurements were higher, but CD levels were lower than physiological values [36]. Pre-dialysis CD acetoacetate levels were higher, but AD resulted in higher post-dialysis levels, as previously reported [8]. Acetoacetate levels were supraphysiological in all measurements [26]. This increase in ketone bodies is meaningful, as it was related to cardiovascular events and all-cause death in a Japanese population study [37].

### 3.5. Amionoacidic Metabolism-Related Metabolites

Glutarate and glutamate decreased during dialysis, although only glutamate reduction when using CD was statistically significant. Pre-dialysis levels of glutarate were higher than physiological levels [26], but glutamate levels were lower [38], reflecting inhibition of the metabolism of amino acids in addition to their high losses seen during sessions [39,40]. Glutamate can also be converted to 2-oxoglutarate by glutamate dehydrogenase, which increased during sessions.

### 3.6. Limitations

One limitation of this study is that we did not measure the effluent metabolome, which could have offered a more accurate approach to the real balance of metabolites as they are cleared with a sieving coefficient of approximately 1 [18]. In addition, we included patients with different vascular access, dialysis modalities, and comorbidities, that together with the non-identical electrolytes of both dialysates, may have biased the analysis. The number of metabolites measured was limited; thus, we may have missed other changes in important metabolic pathways. Finally, the study lacked a wash-out period, as patients had to continue their treatments.

## 4. Materials and Methods

### 4.1. Study Design and Participants

A unicentric, cross-over, prospective study was conducted at the hemodialysis unit of Hospital Universitari i Politècnic La Fe of Valencia, Spain, to analyze the inter-individual pre- and post-dialysis metabolic profile differences between the use of two dialysate acidifiers: Fresenius^®^ ACF 3A5 acting as the acetate dialysate (AD), that contains 4 mmol/L of acetate, and Fresenius^®^ SmartBag CA 211.5 as the citrate one (CD), with 1 mmol/L of citrate (Table 2).

Each DD-CKD patient over 18 years old, prevalent (i.e., for at least three months), with a treatment scheme of four-hour sessions three times per week, was considered for inclusion. Patients were excluded if they had been admitted to the hospital within the previous month, refused to give written informed consent, or were on treatment with a low calcium dialysate (i.e., 1.25 mmol/L).

Every included patient received twelve dialysis sessions with each dialysate. We could not perform a wash-out period, given that dialysis is a life-sustaining treatment. Four blood samples were taken, pre- and post-dialysis on a midweek session after being exposed to each dialysate acidifier for four weeks. Dialysis parameters, medical treatment, and dialysate components, other than the acidifier, remained unchanged during the study to avoid potential confounders. In that sense, and to make the treatment parameters as similar as possible, bicarbonate and sodium concentration in both dialysates were adjusted in the monitor to 32 and 138 mmol/L, respectively. More details on the parameters of the dialysis treatments have been published previously [9].

### 4.2. Variables

Quantitative data from a group of fifteen predetermined intermediate metabolites were collected (Table 2): 3-hydroxybutyrate and acetoacetate to evaluate ketogenesis; citrate, isocitrate, and 2-oxoglutarate to evaluate the first carbon oxidation pathway of the TCA cycle, and succinate, fumarate, and malate to evaluate the second pathway; pyruvate to evaluate glycolysis; lactate to evaluate anaerobic metabolism; acetylcarnitine to evaluate oxidation of fatty acids; glycerol, to evaluate triglyceride metabolism; myo-inositol to evaluate inositol phosphate metabolism; and glutamate and glutarate to evaluate aminoacidic metabolism (Table 3). Demographics and past medical history data were collected from electronic health records.

### 4.3. Sample Preparation and Metabolomic Analysis

Metabolomic analysis of the processed samples was carried out in the Analytical Unit of the Medical Research Institute Hospital La Fe. Blood samples were drawn in ethylenediaminetetraacetic acid (EDTA) tubes, which were processed within 30 min of collection to avoid platelet activation and protein production, as well as the decomposition of thermolabile compounds. Subsequently, they were centrifuged at 1500 g for 10 min at a temperature of 4 °C to separate the cellular fraction from the plasma. The upper phase, corresponding to the latter, was recovered and centrifuged again at 2500 g for another 15 min at 4 °C to eliminate platelets. The upper phase was recovered and aliquoted into cryo-freezing vials in volumes of 100–200 µL for freezing at −80 °C.

For analysis, 50 µL of the plasma samples were extracted, 150 µL of cold methanol was added, and the mixture was allowed to stand for 20 min at 20 °C to facilitate protein precipitation. The samples were centrifuged at 13,000 rpm for 10 min at 4 °C, and the supernatant was recovered in Eppendorf tubes for evaporation in a SpeedVac. The dried sediment was reconstituted in water: methanol (98:2) 0.2% formic acid solution, to which phenylalanine-d5 was added as an internal calibrator.

An Agilent^®^ 6460 triple-quadruple mass spectrometer, using a liquid chromatography-mass spectrometry-based multiple reactions monitoring metabolomics platform optimized for detecting TCA cycle metabolites, was used to analyze the processed samples. A calibration curve was prepared, and the concentrations (ppb or µg/L) were calculated with internal calibration using the internal standard phe-d5. All reagents and chemicals were purchased from Sigma Aldrich.

### 4.4. Statistical Analysis

Quantitative variables are reported with mean and standard deviation when normally distributed, or median and interquartile range when non-normally distributed, according to the Shapiro-Wilk test, while qualitative data are reported as absolute and relative frequencies. Results below the limit of quantification (LQ), only occurring within glycerol, were replaced by LQ/2. Data with no signal, or whose value was zero, were set as non-detected. A repeated-measures ANOVA was conducted to compare the normally distributed studied metabolites. A Greenhouse-Geisser correction was selected if data failed to meet Mauchly’s test of sphericity. Post-hoc analysis with a Bonferroni adjustment was performed to make pairwise comparisons. If data had significant outliers or marked deviations from normality, the Friedman test was chosen as the non-parametric alternative. Post hoc analyses with Wilcoxon signed-rank tests were conducted with a Bonferroni correction applied to make the pairwise comparisons. A 2-sided *p* < 0.05 was considered statistically significant; thus, after the Bonferroni adjustment, the new significance level was stated as 0.0125. Analyses were performed with IBM SPSS^®^ Statistics 26th version and graphics with GraphPad Prism^®^ 8th.

## 5. Conclusions

In conclusion, dialysis has a catabolic effect, and both dialysates modify the major pathways of intermediary metabolism. Still, this study provides some striking data comparing CD against AD that need further study to better understand the biochemical processes that dialysis and the different dialysate options induce in the patient’s metabolism, leading to more personalized treatment prescriptions.

## Figures and Tables

**Figure 1 ijms-23-11693-f001:**
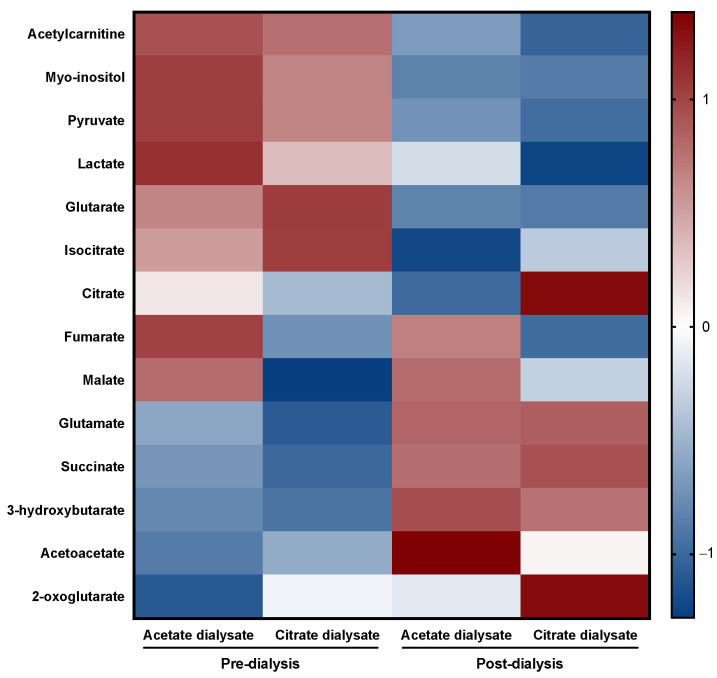
Heat map representing the normalized data by Z-scoring of each metabolite in the four measurement times. It is clear that dialysis per se has a catabolic effect and that most of the differences are produced by the treatment; however, it is visible that, in some cases, the metabolite concentrations are very disparate between dialysates.

**Figure 2 ijms-23-11693-f002:**
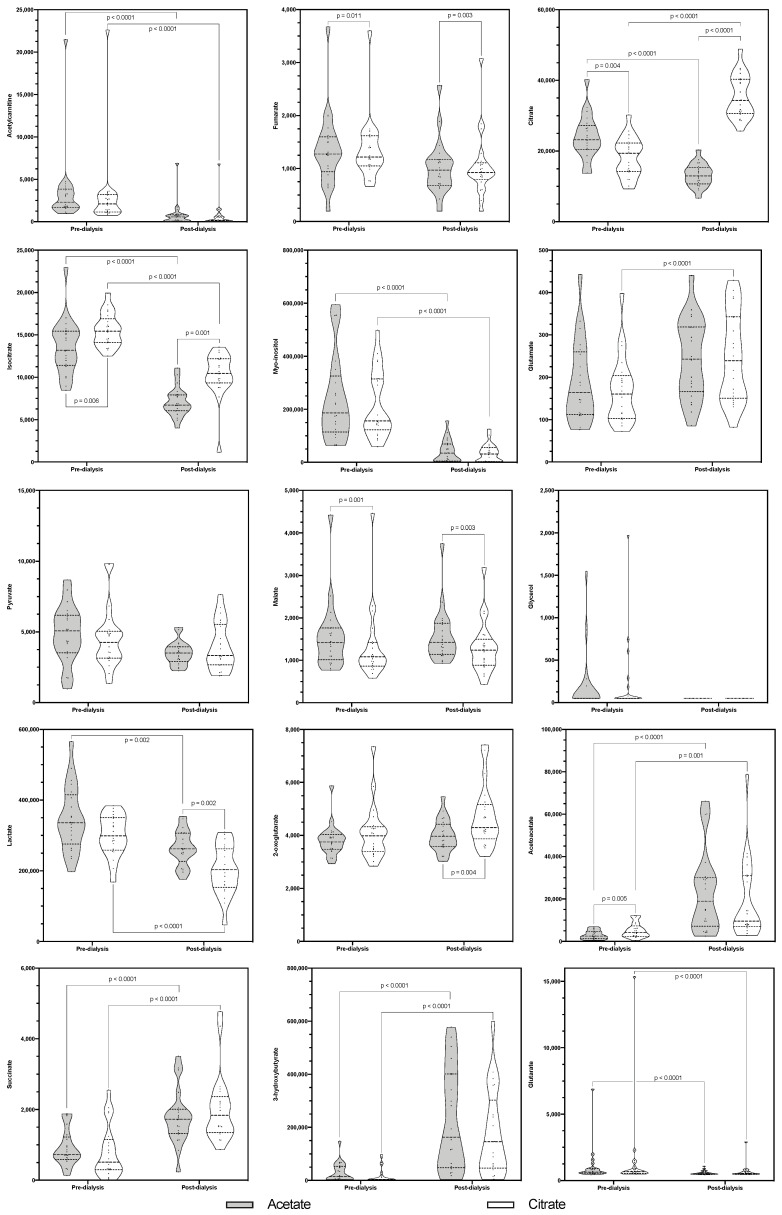
Violin plot representing the pre- and post-dialysis measured concentrations of the fifteen studied metabolites with each dialysate. Statistically significant differences are marked.

**Table 1 ijms-23-11693-t001:** Differences in measured metabolites concentrations expressed in µg/L as mean ± standard deviation or median (interquartile range).

Metabolite (µg/L)	Acetate Dialysate		Citrate Dialysate	
	Pre-Dialysis	Post-Dialysis	Pre-Dialysis	Post-Dialysis
Acetylcarnitine	2292.83 (1698.3–3835.64)	565.12 (75–814.56)	2108.21 (1138.39–3206.01)	164.52 (75–643.1)
Fumarate	1275.33 (940.83–1597.34)	1215.87 (1050.27–1617.92)	968.72 (682.95–1169.75)	924.19 (790.46–1109.45)
Citrate	23,930.01 ± 6243.63	13,151.82 ± 3245.25	18,312.77 ± 5534.11	35,351.74 ± 6126.8
Isocitrate	13,181.79 (11,407.72–15,441.07)	6717.97 (6051.02–7927.15)	15,438.28 (14,109.91–16,918.49)	10,463.95 (9330.3–12,195.79)
Myo-inositol	186,825.86 (115,033.27–325,459.59)	34,635.77 (4328.17–69,297.23)	156,509.77 (123,131.02–315,150.31)	31,586.57 (3125–55,889.43)
Glutamate	164.33 (112.36–260.19)	242.62 (166.44–318.71)	160.35 (102.76–204.39)	239.09 (150.69–342.67)
Pyruvate	5075.38 (3526.28–6168.7)	3509.35 (2905.18–3952.23)	4253.1 (3135.06–5031.68)	3322.45 (2664.53–5529.55)
Malate	1418.21 (1015.39–1760.36)	1418.82 (1138.72–1869.07)	1080.15 (860.11–1416.11)	1238.32 (879.48–1496.05)
Glycerol	48.5 (48.5–48.5)	48.5 (48.5–48.5)	48.5 (48.5–129.6)	48.5 (48.5–48.5)
Lactate	347,331.76 ± 91,933.58	263,698.08 ± 49,573.01	299,899.63 ± 57,695.8	200,315.81 ± 68,510.44
2-oxoglutarate	3755.05 (3476.57–4032.21)	3968.36 (3587–4421.18)	3984.51 (3393.35–4323.3)	4290.49 (3868.34–5168.94)
Acetoacetate	2473.04 (1277.63–4600.98)	18,926.55 (7189.31–30,038.77)	4166.43 (2374.53–7277.45)	9580.02 (6992.68–30,978.53)
Succinate	722.6 (588.57–1221.9)	1725.48 (1325.02–2012.26)	515.06 (297.41–1150.89)	1838.29 (1344.81–2370.32)
3-hydroxybutyrate	14,338.64 (3125–53,089.27)	162,982.81 (48,176.41–401,805.24)	3125 (3125–23,132.58)	146,012.84 (46,441.35–303,043.91)
Glutarate	636.45 (548.76–906.39)	530.86 (480.5–618.66)	664.43 (528.85–922.1)	526.83 (479.83–668.31)

**Table 2 ijms-23-11693-t002:** Dialysate characteristics and compounds.

	Fresenius ACF 3A5	Fresenius Smartbag CA 211.5
Sodium (mmol/L)	140	138
Potassium (mmol/L)	2	2
Calcium (mmol/mL)	1.5	1.5
Magnesium (mmol/mL)	0.5	0.5
Chloride (mmol/mL)	106	109
Acetate (mmol/L)	4	-
Citrate (mmol/L)	-	1
Glucose (g/L)	1	1
Bicarbonate (mmol/L)	35	32
In-use dilution	1 + 44	1 + 44

**Table 3 ijms-23-11693-t003:** Analyzed metabolites and their biochemical characteristics.

Metabolite	Chemical Formula	Molecular Mass (g/mol)	Biochemical Class	Main Metabolic Pathways
Acetylcarnitine	C_9_H_17_NO_4_	203.236	Fatty acid esters	Oxidation of fatty acids
Fumarate	C_4_H_4_O_4_	116.072	Dicarboxylic acids and derivatives	TCA cycle (2nd carbon oxidation), electron transport chain
Citrate	C_6_H_8_O_7_	189.1	Tricarboxylic acids and derivatives	TCA cycle (1st carbon oxidation), transfer of acetyl groups into mitochondria
Isocitrate	C_6_H_8_O_7_	189.1	Tricarboxylic acids and derivatives	TCA cycle (1st carbon oxidation)
Myo-inositol	C_6_H_12_O_6_	180.16	Alcohols and polyols	Inositol phosphate metabolism, secondary messenger in signal transduction pathways
Glutamate	C_5_H_9_NO_4_	147.13	Amino acids, peptides, and analogues	Aminoacidic metabolism
Pyruvate	C_3_H_4_O_3_	88.06	Alpha-keto acids and derivatives	Glycolysis, gluconeogenesis, lipogenesis
Malate	C_4_H_6_O_5_	134.08	Beta hydroxy acids and derivatives	TCA cycle (2nd carbon oxidation), gluconeogenesis, pyruvate metabolism
Glycerol	C_3_H_8_O_3_	92.09	Carbohydrates and carbohydrate conjugates	Triglyceride metabolism
Lactate	C_3_H_6_O_3_	90.08	Alpha hydroxy acids and derivatives	Gluconeogenesis, pyruvate metabolism
2-oxoglutarate	C_5_H_6_O_5_	146.11	Gamma-keto acids and derivatives	TCA cycle (1st carbon oxidation), aminoacidic metabolism
Acetoacetate	C_4_H_6_O_3_	101.08	Short-chain keto acids and derivatives	Ketone body metabolism, fatty acid biosynthesis
Succinate	C_4_H_6_O_4_	118.09	Dicarboxylic acids and derivatives	TCA cycle (2nd carbon oxidation), electron transport chain
3-hydroxybutyrate	C_4_H_8_O_3_	103.1	Beta hydroxy acids and derivatives	Ketone body metabolism, fatty acid biosynthesis
Glutarate	C_5_H_8_O_4_	147.13	Dicarboxylic acids and derivatives	Aminoacidic metabolism
TCA: Krebs tricarboxylic acid.				

## Data Availability

The data supporting the findings of this study are available on GitHub (https://github.com/Broseta/Citrate-dialysate.git, (accessed 29 July 2022)).

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
