# Peer review of "Impact of Acetate versus Citrate Dialysates on Intermediary Metabolism—A Targeted Metabolomics Approach"

_ijms, 2022, doi:10.3390/ijms231911693_

Round 1

Reviewer 1 Report

- Originality/Novelty: The proposed paper is an original one, a subject quite new and interesting which can offer new opportunities in the very next future. The idea of acetate "toxicity" is not new, in the '90-s we changed progressively from acetate dialysis to bicarbonate dialysis solutions. The small amounts of acetate still present in the actual dialysis solutions can sometimes be deleterious to the patients. The idea of interference with the Krebs cycle is interesting and proofing it it's quite difficult. This study opens a new window to the biochemical modifications induced by the less compatible dialysis solutions.

 - Significance: The results are interpreted appropriately and they look significant. Some modifications are as expected in regard to the human physiology. The discussions are well done and supported by the results. The hypotheses of acetate toxicity are sustained by these results.

 - Quality of Presentation: The article written in an appropriate way, even if the structure is inversed with material & methodology at the end of the article.  Data are presented appropriately in a very clear and intuitive way, using the highest standards for presentation of the results.

 - Scientific Soundness: I think that the study is correctly designed and analyses are well done using the appropriate statistical tools. Data are robust enough to draw a picture of the events and to draw some conclusions. Some details regarding the methods and solutions are described somewhere else, an article published earlier by the authors but this is not something unusual or e something that can change the significance of this paper.

Author Response

We deeply appreciate your kind comments on our work. We are pleased that it has been to your liking and that you see the same interest that we see in the subject. Thank you. 

Reviewer 2 Report

This is an interesting study about comparing the difference between dialysates with acetate and citrate.

21 patients complete the study with cross over design and 15 metabolites were measured.

The acid-base buffer may interfere with the metabolate passways of glucose amino acid and lipid during dialysis. It is an important issue for patients under dialysis. However, before it can be published,  several points may be clarified.

1. With cross over design, how long is the wash out phase between shift from one dialysate to another? if there was no wash-out phase, how can you be sure there was no lasting effect of the previous dialysate?

2. The two dialysate are different in Na, Ci and HCO3 by Table 2.

How can we be sure that the effect was caused by the difference solely by replace acetate with citrate (and not by the electrolyte difference?)

3. How to interpret Figure 1? those blue blocks seems be changed to red after dialysis.

   How to explain this result to our readers and what is the meaning of this heat map?  This can be illustrated more in the discussion.

4.In Table 1, almost all the values are not normally distributed, so displayed in quartile range and violin plot is quite reasonable. However, the decimal "." should not be replaced by ", and same rule should be applied to the Z and p value.

5. Please rearrange the order of your figure for the metabolite (and labelled with A, B, C,...)

according to the order you mentioned in the results. So it may be easier for our reader to read.

The figure legends may explain more detail about the figure. ( ex: The current figure legend for figure 1 and 3 are the same and may cause confusion.)

Author Response

Thank you very much for your comments. Below we address your concerns point-by-point and have attached the due modifications in a new manuscript version. 

1. With cross-over design, how long is the wash-out phase between shift from one dialysate to another? if there was no wash-out phase, how can you be sure there was no lasting effect of the previous dialysate?

Authors' response: As the reviewer rightly comments, the previously submitted manuscript did not clearly state the wash-out period, which is common in cross-over studies. However, dialysis patients need to continue their three weekly treatment sessions. It would have been much more interesting to have had the opportunity to use a dialysis fluid free of acetate and citrate, such as Medtronic's Lympha fluid (which replaces them with a higher concentration of hydrogen ions), but this was not available. Accordingly, we have added a clarification in this new version on the impossibility of performing a wash-out period in the methodology section: "We could not perform a wash-out period, given that dialysis is a life-sustaining treatment." 

2. The two dialysate are different in Na, Ci and HCO3 by Table 2.
How can we be sure that the effect was caused by the difference solely by replace acetate with citrate (and not by the electrolyte difference?)

Authors' response: Thank you for your comment. We fully agree with you and recognize that we have not clarified it sufficiently in the text. But we did correct the treatment parameters as far as the dialysis monitors allow: leaving bath bicarbonate of 32 mmol/L and sodium of 138 mmol/L in any given dialysate. This modification, however, cannot be done for chloride. We have clarified the text accordingly: "In that sense, and to make the treatment parameters as similar as possible, bicarbonate and sodium concentration in both dialysates were adjusted in the monitor to 32 and 138 mmol/L, respectively." 

3. How to interpret Figure 1? those blue blocks seems be changed to red after dialysis.
  How to explain this result to our readers and what is the meaning of this heat map? This can be illustrated more in the discussion.

Authors' response: Thank you for your comment. What the reviewer rightly points out is part of the article's conclusion: "... dialysis has a catabolic effect per se, and both dialysates modify the major pathways of intermediary metabolism". This heat map reinforces the idea that most of the changes are due to the catabolic effect of the treatment. However, it also, in a very visual way, marks the differences between metabolites at all four times. Note, for example, the big difference between pre-dialysis malate, post-dialysis citrate, or fumarate in all four times. We have added the interpretation of the figure to its legend: "Heat map representing the normalized data by Z-scoring of each metabolite in the four measurement times. It is clear that dialysis per se has a catabolic effect and that most of the differences are produced by the treatment; however, it is visible that, in some cases, the metabolite concentrations are very disparate between dialysates".  

4.In Table 1, almost all the values are not normally distributed, so displayed in quartile range and violin plot is quite reasonable. However, the decimal "." should not be replaced by ", and same rule should be applied to the Z and p value.

Author's response: We appreciate your suggestion and have replaced commas for periods throughout the manuscript. 

5. Please rearrange the order of your figure for the metabolite (and labelled with A, B, C,...) according to the order you mentioned in the results. So it may be easier for our reader to read.The figure legends may explain more detail about the figure. ( ex: The current figure legend for figure 1 and 3 are the same and may cause confusion.)

Author's response: Thank you for your recommendations. We agree that the change in the order of the figures improves the manuscript's readability; hence, we have restructured and grouped the 15 metabolites into one figure. And as suggested, we have expanded the figure legend to make it self-explanatory: "Violin plot representing the pre- and post-dialysis measured concentrations of the fifteen studied metabolites with each dialysate. Statistically significant differences have been marked."

Round 2

Reviewer 2 Report

Minor revision is needed.

1. the limitation should mention the following two points at page 8

 a. lack of a washout period

 b. the electrolyte is not totally identical besides different buffer acid.

2. The 15 metabolite tested in the table 1 and figure 1, 3-hydroxybutyrate is missing.

   Why? is it already be mentioned somewhere?

3. The discussion may be arranged by one of the following method.

  a. grouping for these metabolites first and have subtitles for further discussion.

      ex: TCA cycle related metabolites 

      and then fatty acid oxidation (as for carnitine) 

  b. discussed in the order listed in table 1.

Author Response

Thank you for your new kind comments. 

1. the limitation should mention the following two points at page 8
 a. lack of a washout period
 b. the electrolyte is not totally identical besides different buffer acid.

Authors' response: These limitations have been included as suggested: "One limitation of this study is that we did not measure the effluent metabolome, which could have offered a more accurate approach to their real balance as they are cleared with a sieving coefficient of approximately 1[18]. In addition, we included patients with different vascular access, dialysis modalities, and comorbidities, which, together with the non-identical electrolytes of both dialysates, may have biased the analysis. Also, the number of metabolites measured was limited; thus, we may have missed other changes in important metabolic routes. Finally, the study lacks a wash-out period, as patients had to continue their treatments." 

2. The 15 metabolite tested in the table 1 and figure 1, 3-hydroxybutyrate is missing.
  Why? is it already be mentioned somewhere?

Authors' response: Thank you very much for pointing out this important issue. In one of the editions of the manuscript, we may have mistakenly deleted the information on this metabolite in the results section, but it has now been corrected.   

3. The discussion may be arranged by one of the following method.
 a. grouping for these metabolites first and have subtitles for further discussion.
   ex: TCA cycle related metabolites 
   and then fatty acid oxidation (as for carnitine) 
 b. discussed in the order listed in table 1.

Authors' response: Thank you for this suggestion. We have opted for option a and recognize that it improves the understanding of the manuscript.